# HUNTING GAMES

## ABSTRACT

Markov Decision Processes (MDPs) address sequential decision-making under stochastic dynamics, where an agent selects actions, observes transitions, and aims to maximize rewards. Traditional reinforcement learning (RL) approaches assume a reasonably accurate estimate of the operating region in the state space. However, such an assumption rarely holds in real-world domains such as counter-drone defense and algorithmic trading, which feature environments whose limits of operation are only revealed gradually through interaction. As a result, the stochastic dynamics may push the agent into unfamiliar regions, where incomplete knowledge leads to suboptimal actions and reduced reward accumulation. This paper formulates this new phenomenon as a *hunting game* between the agent (*hunter*) and the environment (*target*). Its key motivation is that environments with heavy-tailed variability introduce rare but impactful surprises that slow down learning and act as implicit defenses, even without explicit adversarial presence. Despite its practical relevance, this setting remains poorly understood. In this paper, we analyze the theoretical limits of such hunting games in a model-based RL framework. Our work reveals that the difficulty of learning is governed by the novelty encountered by the agent, weighted by the eluder dimension of the environment's true model class. Reducing either factor shifts the balance in favor of the agent.

## 1 INTRODUCTION

A plethora of real-world control problems entail an agent learning while the "operational envelope" of the environment is being revealed over time. For example, in cyber defense and counter-drone operations, coarse and event-driven interventions are interleaved with long intervals of unobserved evolution (National Security Agency (NSA), 2019; Mandiant Consulting, 2025; Seidaliyeva et al., 2023; Director, Operational Test & Evaluation (DOT&E), 2020). In such cases, the next observed state can jump "far" from where a conventional model expects it to be. Similar patterns happen in algorithmic trading during regime shifts (Xu et al., 2015; Ames et al., 2017; Koolen et al., 2012) and in safety-critical robotics when sudden shocks perturb the dynamics (Ang & Timmermann, 2011; Guidolin, 2012; Truong et al., 2020; Lo, 2004). In all of these cases, the environment is not explicitly adversarial, yet it behaves as if it were protected by "implicit defenses" – in other words, sporadic yet high-impact transitions may influence the learning process to a significant extent (Mao et al., 2021; Cheung et al., 2020; DiGiovanni & Tewari, 2021; Huang et al., 2023; Zhuang & Sui, 2021a).

To the best of our knowledge, this setting has not been formalized and studied yet. As such, this paper introduces the new concept of *hunting game*. The game takes place between a *hunter* (i.e., the learning agent) and a *target* (i.e., the environment). The target follows stationary but unknown dynamics, while the hunter acts periodically, collects rewards, and gradually infers both rewards and transitions. Two features distinguish this setting. First, the agent is continually pushed into previously unseen regions of state space. Second, since the effective step size between consecutive decisions from the agent, can be large with respect to the system's underlying natural time scale, the subset of states the target environment can inhabit expands over time. Mathematically this gets reflected in the expected value of the next-state norm. To capture these effects we assume a slowly growing bound on the means of rewards and next states via growth functions. By using a model-based analysis based on reinforcement learning centered on posterior sampling, we show that the expected regret is controlled by (a) how much novelty the agent encounters and (b) the eluder dimension of the true reward/transition classes.

Prior work does not entirely capture the proposed *hunting games*. Existing algorithms achieve sublinear regret by controlling uniform uncertainty, while structure-aware analyses replace dependence on the state space with complexity measures such as eluder dimension and linear function approximation (Kearns & Singh, 2002; Brafman & Tennenholtz, 2002; Strehl & Littman, 2008; Jaksch et al., 2010; Azar et al., 2017; Osband et al., 2013b; Osband & Van Roy, 2014; Jin et al., 2021; 2020; Wen & Van Roy, 2017). While other prior work has treated adversarial ambiguity or heavy-tailed noise via robust estimators, it does not model externally forced growth of the explored support (Iyengar, 2005; Nilim & El Ghaoui, 2005; Bubeck et al., 2013; Zhuang & Sui, 2021b). Conversely, we formalize *induced, growth-driven experience* through growth functions $g_R(t), g_P(t)$ that bound how reward and next-state means expand between decision epochs under sub-Gaussian noise, capturing heavy-tailed effects without the need to formalize adversaries.

Our analysis shows trajectory-aware regret bounds are based on set widths along the realized path. This when combined with growth-weighted scaling with eluder dimensions, reveal when classical optimism/PSRL guarantees are misaligned. Precisely in open-world control and provide actionable levers (e.g., decision-epoch frequency or operational constraints) to regain sublinear regret (Osband & Van Roy, 2014; Osband et al., 2013b; Azar et al., 2017). Conceptually, our results connect: (1) *structure-aware exploration* where efficiency scales with function-class complexity rather than raw state/action cardinality; and (2) *open-world learning*, where the set of relevant states effectively expands during interaction. On the exploration side, posterior sampling for RL (in short, PSRL) and optimistic algorithms are known to admit near-minimax regret in tabular settings and to extend to rich models by tying regret to complexity measures (e.g., the eluder dimension). By importing these ideas into growth-controlled, heavy-tailed settings, we reveal *when* and *how* exploration impacts learning.

**Contributions.** First, we introduce the *hunting game* formalism and a growth-controlled MDP family. Second, we provide a regret analysis for PSRL that (i) decomposes error into set widths evaluated along true trajectories and (ii) upper-bounds the total regret with high confidence, by terms that scale with the eluder dimensions of the reward and transition classes, weighted by the growth functions. Third, we show qualitatively that bounding novelty via the growth functions or lowering eluder dimensions yields sublinear regret. Ultimately, this provides modeling choices and operational constraints to make hunting games learnable.

## 2 RELATED WORK

**Efficient exploration in MDPs.** Algorithms such as $E^3$ (Kearns & Singh, 2002), R-MAX (Brafman & Tennenholtz, 2002), MBIE-EB (Strehl & Littman, 2008), and UCRL2 (Jaksch et al., 2010) guarantee polynomial sample complexity or sublinear regret in tabular MDPs. $E^3$ and R-MAX use optimistic models to drive exploration, while MBIE-EB refines this with interval estimates. UCRL2 attains $\tilde{O}(D\,|S|\sqrt{|A|T})$ regret in average-reward MDPs (with diameter $D$). Minimax-optimal rates for finite-horizon tabular RL were later obtained via UCBVI-style analyses (Azar et al., 2017). PSRL provides clean analyses and strong empirical performance, with near-state-of-the-art bounds in finite MDPs and conceptually simple algorithms: sample an MDP from the posterior each episode and execute its optimal policy (Osband et al., 2013b). Extensions achieve worst-case guarantees under communicating MDP assumptions. Our analysis derives regret in terms of growth-weighted widths, specialized to environments that induce significant exploration between decision epochs.

**Function approximation and complexity measures.** In addition to tabular settings, a central theme is to replace dependence on the state space with complexity measures of the function classes used to model rewards, transitions, or value functions. The *eluder dimension* (Russo & Van Roy, 2013b;a) first arose in bandits to quantify how many "independent" observations are needed to resolve a function class. Osband & Van Roy further extended this notion to *model-based RL*, tying PSRL regret to the eluder (and Kolmogorov) dimensions of reward/transition classes (Osband & Van Roy, 2014). Related developments introduced Bellman Eluder(BE)-dimension and other measures for rich function approximation and linear MDPs, yielding polynomial-time, near-optimal guarantees (Jin et al., 2021; 2020; Wen & Van Roy, 2017). While our work assumes this perspective, it shows that even with favorable function-class complexity, *learning can be heavily influenced by environment-induced exploration*, captured through growth functions in the proposed hunting game formalism.

**Intrinsic motivation.** Prior work has proposed count-based and curiosity-driven bonuses to promote discovery of novel states, e.g., pseudo-counts, hashing, and prediction-error signals (Bellemare et al., 2016; Tang et al., 2016; Pathak et al., 2017). While these methods target practical exploration with high-dimensional observations, they typically presume a *fixed* environment support. Our setting is complementary: *exploration is not optional*; the environment's heavy-tailed variability forces visits to expanding regions, and our bounds reflect how such exploration interacts with learning complexity.

**Heavy-tailed noise and robustness.** Robust MDPs address *adversarial* or *ambiguity-set* uncertainty while optimizing for worst-case models (Iyengar, 2005; Nilim & El Ghaoui, 2005). Heavy-tailed reward models, in addition to bandits, motivate robust estimators and modified UCB/Thompson strategies (Bubeck et al., 2013). In stark opposition, our target is not adversarial. While our rewards/transitions satisfy sub-Gaussian noise, they allow *growth-controlled means* that produce sudden transitions. Ultimately, this results in a novel regret decomposition happening via widths along the true trajectory, which emphasizes how *exploration × eluder-dimension* drives learning.

**Summary of Novelty.** To the best of our knowledge, prior work either (i) measures exploration difficulty by global complexity of the hypothesis class (Osband & Van Roy, 2014; Jin et al., 2021; 2020; Wen & Van Roy, 2017) or (ii) secures worst-case guarantees via optimism/posterior sampling without modeling externally forced exploration (Kearns & Singh, 2002; Brafman & Tennenholtz, 2002; Strehl & Littman, 2008; Jaksch et al., 2010; Azar et al., 2017; Osband et al., 2013b). As a consequence, we make three novel foundational advances:

(1) **Formalizing forced exploration.** The *hunting game* introduces growth functions $g_R(t), g_P(t)$ that explicitly bound how rewards and next-state means can expand between decision epochs. This is different from robust MDPs which assume adversarial ambiguity sets (Iyengar, 2005; Nilim & El Ghaoui, 2005) and from heavy-tail models that modify noise assumptions (Bubeck et al., 2013; Zhuang & Sui, 2021b). In hunting games, the environment is not adversarial and the noise is light-tailed, yet the key challenge is that the exploration grows over time.

(2) **Trajectory-aware complexity.** Whereas classical regret bounds control uncertainty uniformly over the state-action space (Jaksch et al., 2010; Azar et al., 2017; Osband et al., 2013b), our regret is driven by set widths evaluated only along the realized trajectory, which ultimately reveals how the amount of exploration actually encountered influences learning.

(3) **Growth-weighted eluder scaling.** Previous structure-aware results tie regret to eluder- or BE-dimensions of reward/transition classes (Osband & Van Roy, 2014; Jin et al., 2021; 2020; Wen & Van Roy, 2017). In this paper, we extend this notion by showing that regret scales with these dimensions weighted by $g_R, g_P$. Thus, even with benign function classes, learning can be slow if exploration expands quickly. Conversely, bounding growth makes hunting games learnable.

Collectively, these advances identify a previously unexplored setting that is neither adversarial nor stationary – in other words, agents face *forced, growth-driven exploration*. Our new theory studies how this exploration couples with class complexity to determine regret and clarifies when classical optimism/PSRL guarantees are pessimistic or overly optimistic in open-world learning settings.

## 3 PROBLEM FORMULATION: HUNTING GAMES

We begin with the set up of the learning problem as an interactive exercise between a hunting agent A and a dynamic target represented by the environment M. It proceeds as repeated interactions over time. At each time instant, the hunting agent A takes an action on M, and receives a scalar reward. This action pushes the target M to react, by changing its state according to some unknown but stationary transition distribution. The hunter agent does not have prior knowledge of the reward function, or the internal model which dictates the evolution of M's state. However, through repeated engagements in this interactive process, it improves its understanding of the transition function of M and maximizes the collected reward. In such settings, the change in the environment state during a transition can be substantial. This occurs because the agent's notion of a discrete MDP time step does not necessarily align with the finer sampling rates common in robotics or similar settings. Control or attack decisions are typically updated at coarser temporal resolutions, resembling decision epochs. Consequently, the underlying system may evolve considerably between two decision updates, leading to large effective state transitions and inducing high variance in the dynamics observed

by the agent. More formally, the target is a random finite horizon MDP $M := \langle \mathcal{S}, \mathcal{A}, R, P, \tau, \rho \rangle$. $\mathcal{S}$ is the state-space, $\mathcal{A}$ is the set of actions, $R : \mathcal{S} \times \mathcal{A} \to \Delta((0, 1])$ is the reward distribution for a state-action pair for the hunter agent. $P(\cdot|s, a)$ is the transition distribution over $\mathcal{S}$ under the influence of action $a$ in state $s^1$. $\tau$ is the time horizon of these episodic interactions and $\rho$ is the initial state distribution.

The hunter agent A 's policy $\mu$ is a function which maps a state $s \in \mathcal{S}$ and $i = 1, \ldots, \tau$ to an action $a \in \mathcal{A}$. For each MDP M and policy $\mu$ we define the value function $V$:

$$V_{\mu,i}^{M}(s) := \mathbb{E}_{M,\mu}\bigg[ \sum_{j=i}^{\tau} \bar{r}^{M}(s_j, a_j)\bigg| s_i = s \bigg].$$

Here, $\bar{r}^{M}(s, a) := \mathbb{E}[r|r \sim R(s, a)] \equiv \mathbb{E}_{r \sim R(s,a)}[r]$. The subscripts in the expectation operator means that the subsequent actions from hunter A continue according to $a_j = \mu(s_j, j)$, and the target M continues to evolve according to the transition function $s_{j+1} \sim P(\cdot|s_j, a_j)$, for $j = i, \ldots, \tau$. A policy $\mu^{S,M}$ is said to be optimal for MDP $M$ on a subset $S \subseteq \mathcal{S}$ if $V_{\mu,i}(s) = \max_{\mu'} V_{\mu',i}(s)$, for all $s \in S$ and $i = 1, \ldots, \tau$. When, the policy $\mu^{S,M}$ is optimal on the whole state-space of the MDP $M$ we simply denote it as $\mu^{M}$.

The interactive setting described above captures a setting where the target M is substantially weaker, in the sense that it cannot cause damage to the hunter agent directly. It is instead motivated by survival. It is advantaged by the lack of understanding on part of the attacker about its state transition function. Even though the hunter has the ability to launch attacks (take actions), it faces two primary challenges. It lacks an understanding of (1) the lethality of an attack in different states - *unknown reward*, and (2) how the attacks causes the target to change states - *unknown transition probabilities*. The hunter thus relies on a learning-based iterative approach to uncover these. This gradual incremental improvement of the hunter's efficacy, bringing it closer to successfully accomplishing its objective against the target.

The reinforcement learning (RL) agent A interacts in an episodic fashion at times $t_k = (k-1)\tau + 1$, $k = 1, 2, \ldots$ over time. Note that the agent does not know apriori the set of states in $\mathcal{S}$ it can find the target M in, but discovers it as the interactions proceed. In practice, this could mean a growth of the performance envelope of the target M . We denote a finite history $H_t = (s_1, a_1, r_1, \ldots, s_{t-1}, a_{t-1}, r_{t-1})$ as the sequence of observations made prior to time $t$. Additionally, we denote $\mathcal{S}_k$ as the subset such that $s_{i<t} \in \mathcal{S}_k \subset \mathcal{S}$. Note that, for any episode $k$, $\mathcal{S}_{k-1} \subseteq \mathcal{S}_k$. For any set $X$ and $Y$ in $\mathbb{R}^d$, let $\mathcal{P}_{C,\sigma}^{X,Y}$ be the family of distributions from $X$ to $Y$ with $l_2$-bounded mean in $[0, C]$ and additive $\sigma$-sub-Gaussian noise. Let $g : \mathbb{N} \to \mathbb{R}_+$ be a non-decreasing growth function controlling the allowed mean at time $t$. Then, we define the family $\mathcal{P}_{g,\sigma}^{X,Y} := \{\mathbb{D}(\cdot|x) : x \in X, \|\mathbb{E}_{y \sim \mathbb{D}(\cdot|x)}[y]\|_2 \le g(t), \text{ and } y - \mathbb{E}[y \mid x] \text{ is } \sigma\text{-sub-Gaussian}\}$ .

**Assumption 3.1** (Bounded Growth). We need the state space $S \subset \mathbb{R}^d$, for some finite dimension $d$. This should not impose a great restriction for most practical environments. We assume function classes $\mathcal{R} \subset \mathcal{P}_{g_R,\sigma_R}^{\mathcal{S} \times \mathcal{A}, \mathbb{R}}$, and $\mathcal{P} \subset \mathcal{P}_{g_P,\sigma_P}^{\mathcal{S} \times \mathcal{A}, \mathcal{S}}$ for reward and transition respectively. Here, $g_R(t), g_P(t)$ are slowly growing functions controlling the maximum expected reward and next-state norm at time step $t$. Then, for $\sigma_R, \sigma_P > 0$, the MDP $M^*$ has $R^* \in \mathcal{R}, P^* \in \mathcal{P}$, and the transitions satisfy a time-dependent mean bound: $\forall t, \|\mathbb{E}[s_{t+1} \mid s_t, a_t]\|_2 \le g_P(t)$ with additive $\sigma$-sub-Gaussian noise.

We align the problem formulation along previous work on analyzing model-based RL (Osband & Van Roy, 2014; Russo & Van Roy, 2013a). The RL algorithm produces a deterministic sequence $\{\pi_k : k = 1, 2, \ldots\}$, where each $\pi_k$ is a distribution over policies. The agent A will employ one such policy during the $k^{\text{th}}$ episode. The regret incurred by the RL algorithm $\pi$ up to time $T$ is expressed as the following:

---

[1]Note, that the set $\mathcal{A}$ can include a *leave alone* action which models the autonomous mode of the target's behavior.

$$\text{Regret}(T, \pi, M^*) := \sum_{k=1}^{\lceil T/\tau \rceil} \Delta_k, \tag{1}$$

where $\Delta_k$ denotes the regret over the $k^{\text{th}}$ episode, defined with respect to the MDP $M^*$ by

$$\Delta_k := \int_{s_0 \in \mathcal{S}_k} \rho(s_0) \left( V_{\mu^*,1}^{M^*} - V_{\mu_k,1}^{M^*} \right)(s_0) \mathrm{d}s_0 \tag{2}$$

where $\mu^* = \mu^{M^*}$ and $\mu_k \sim \pi_k(H_{t_k})$. It should be clear that regret is not deterministic.

We outline the efficient Posterior Sampling for Reinforcement Learning Algorithm (PSRL) next. PSRL satisfies efficient regret bounds for finite horizon MDPs (Osband et al., 2013a). It functions in the following fashion. At the start of each episode, it samples an MDP $M_k$ according to the history until that point $H_{t_k}$. This model is used to compute a policy $\mu_{M_k}$, which is optimized for the set of states $S_k$ in MDP $M_k$ during episode $k$. Similarly, $\mu_{M_k}^*$ is the policy optimized on the subset $S_k$ in the original MDP $M^*$.

---

**Algorithm 1** `Posterior Sampling Algorithm`

---

    **Input:** Prior distribution $\phi$ for $M^*$, $t = 1$
1: **for** episodes $k = 1, 2, \ldots$ **do**
2:     Sample $M_k \sim \phi(\cdot | H_t)$
3:     Compute $\mu_k = \mu^{M_k}$
4:     **for** timesteps $j = 1, \ldots, \tau$ **do**
5:         attack with $a_t \sim \mu_k(s_t, j)$
6:         observe reward $r_t$
7:         target advances $s_{t+1} \sim P(\cdot | s_t, a_t)$, $t = t + 1$
8:     **end for**
9: **end for**

---

*Under conditions of bounded apparent state-space and reward growth given by $g_T(t), g_R(t)$, how does the regret (outlined in Equation 1) evolve over time ?* We answer this question in the remainder of this paper and present our analysis.

## 4 ANALYSIS

The regret in Equation 1, is upper bounded by the following expression :

$$\mathbb{E}[\text{Regret}(T, \pi^{PS}, M^*)] \leq [g_{\mathcal{R}}(t_m) + g_T(t_m)] + R^w + \mathbb{E}[K^*]\left(1 + \frac{1}{T-1}\right)P^w,$$

$$R^w = 1 + 2\tau g_{\mathcal{R}}(t_m)\dim_E(\mathcal{R}_m, T^{-1}) + 4\sqrt{\beta^*(1/8T)\dim_E(\mathcal{R}_{t_m}, T^{-1})T},$$

$$P^w = 1 + 2\tau g_T(t_m)\dim_E(P_m, T^{-1}) + 4\sqrt{\beta^*(1/8T)\dim_E(\mathcal{P}_{t_m}, T^{-1})T}. \tag{3}$$

**Implications and Discussion.** The width quantities $R^w$ and $P^w$ bound the maximum disagreement between the real model (reward and transition function) and the estimated model. In the expressions of $R^w$ and $P^w$, notice that the eluder dimension of the estimated function class $\dim_E(\cdot, T^{-1})$ is scaled by the growth factor at the latest episode $m$. Note, that the estimated function classes ($\mathcal{R}_m$ and $\mathcal{P}_m$) depends only on the data up to episode $m-1$ (Section 7). Therefore, the growth function $g(\cdot)$ can have a significant impact on the learning regret. We discuss the qualitative aspects next.

From the perspective of the hunter A , the game is worth playing precisely when regret can be bounded, and futile otherwise. For this two necessary conditions must be satisfied, (1) the eluder dimensions of the reward and the transition functions of the target M must be finite; and (2) the growth function $g(t)$ must be bounded. Equivalently, a necessary condition for the target to remain

unlearnable is that its dynamics either belong to a hypothesis class of sufficiently high complexity (as quantified by the eluder dimension), or that it continually induces novelty by visiting unexplored regions of the state space $\mathcal{S}$. This condition corresponds to the agent visiting parts of the state space with vanishing reward density, thereby jeopardizing task completion. For readers less familiar with the notion of eluder dimension, note that if the elements of the function class have a finite Lipschitz constant, then their eluder dimension is also finite at any fixed precision $\epsilon$.

## 5 MODEL BASED RL ANALYSIS

We restate the regret outlined in Equation 2 here:

$$\Delta_k := \int_{s \in \mathcal{S}_k} \left( V_{\mu^*,1}^{M^*} - V_{\mu_k,1}^{M^*} \right)(s)\rho(s)\mathrm{d}s$$

and the total regret from Equation 1, $\text{Regret}(T, \pi, M^*) := \sum_{k=1}^{\lceil T/\tau \rceil} \Delta_k$.

We begin by recalling some basic definitions. Let $\sigma(H_{t_k})$ denote the sigma-algebra generated by the history up to time $t_k$. Intuitively, $\sigma(H_{t_k})$ represents the collection of all events that are observable to the agent at that time. A random variable $X$, taking values in a measurable space $\mathcal{X}$, is said to be $\sigma(H_{t_k})$-measurable if $X : \mathcal{X} \to \mathbb{R}$ for every Borel set $B \subseteq \mathbb{R}$, $\{\omega \in \mathcal{X} : X(\omega) \in B\} \in \sigma(H_{t_k})$. Read backwards, pick any Borel set $B$. Collect all the outcomes in $\mathcal{X}$ which causes $X$ to take values in $B$. This set is contained in the sigma algebra $\sigma(H_{t_k})$. We state a simple observation: at the start of each $k^{\text{th}}$ episode, $M^*$ and $M_k$ are identically distributed. This lets us relate functions which depend on the true but unknown MDP $M^*$ to the sampled MDP $M_k$ which is observed by the agent. Therefore, the following Lemma is an immediate consequence.

**Lemma 5.1** (Posterior sampling). *If $\phi$ is the distribution of $M^*$ then, for any $\sigma(H_{t_k})$-measurable function $g$,*

$$\mathbb{E}[g(M^*)|H_{t_k}] = \mathbb{E}[g(M_k)|H_{t_k}].$$

Note that the difficulty of computing the regret $\Delta_k$ is that we do not observe the optimal policy $\mu^*$. For many RL algorithms, there is no clean way to relate the unknown optimal policy to the states and actions the agent actually observes. The following results shows how we can avoid this issue.

We state the Bellman operator $\mathcal{T}_{\mu}^{M}$, which for any MDP $M = \langle \mathcal{S}, \mathcal{A}, R^M, P^M, \tau, \rho \rangle$, stationary policy $\mu : \mathcal{S} \to \mathcal{A}$, and value function $V : \mathcal{S} \to \mathbb{R}$ is given by the following:

$$\mathcal{T}_{\mu}^{M} V(s) := \bar{r}^M(s, \mu(s)) + \int_{s' \in \mathcal{S}} P^M(s'|s, \mu(s))V(s')\mathrm{d}s'.$$

This gives the dynamic programming equations.

**Definition 5.2** (Dynamic Programming). *For any MDP M = $\langle \mathcal{S}, \mathcal{A}, R^M, P^M, \tau, \rho \rangle$ and policy $\mu : \mathcal{S} \times \{1, \ldots, \tau\} \to \mathcal{A}$, the value functions $V_{\mu}^M$ satisfy*

$$V_{\mu,i}^M = \mathcal{T}_{\mu(\cdot,i)}^M V_{\mu,i+1}^M$$

*for $i = 1, \ldots, \tau$ with $V_{\mu,\tau+1}^M := 0$*

The episodic regret under dynamic programming is given by the following expression,

$$\mathbb{E}[\Delta_k] = \sum_{i=1}^{\tau} \mathbb{E}\left[ \left( (T_{\mu_k}^{M_k} - T_{\mu_k}^{M^*}) V_{\mu_k,i+1}^{M_k} \right)(s_{t_k+i}) \right], \tag{4}$$

i.e., the expected episode-$k$ regret equals the cumulative Bellman error of the executed policy $\mu_k$ evaluated along the true trajectory. We refer the reader to Appendix A.1 for this analysis.

For any distribution $\Phi$ over $\mathcal{S}$, let us define the following.

**Definition 5.3** (One Step Future Value Function). We define one step future value function $U$ to be the expected value of the optimal policy with the next state distributed according to $\Phi$.

$$U_i^M(\Phi) := \mathbb{E}_{M,\mu^M}\left[V_{\mu^M,i+1}^M(s) \mid s \sim \Phi\right] \tag{5}$$

Additionally, we expect a regularity condition that the future values of similar distributions should be similar. First we write the mean of a distribution $\Phi$ to be $\mathcal{E}(\Phi) := \mathbb{E}[s \mid s \in \Phi] \in \mathcal{S}$. This lets us express the Lipschitz continuity for $U_i^M$ with respect to the $\|\cdot\|_2$-norm of the mean as:

$$|U_i^M(\Phi) - U_i^M(\bar{\Phi})| \leq K_i^M(\mathsf{Q})\|\mathcal{E}(\Phi) - \mathcal{E}(\bar{\Phi})\|_2, \qquad \text{for all } \Phi, \bar{\Phi} \in \mathsf{Q}.$$

We define $K^M(\mathsf{Q}) := \max_i K_i^M(\mathsf{Q})$ to be the global Lipschitz constant for the future value function for the state distributions from $\mathsf{Q}$. Where appropriate, we will condense our notation to write $K^M := K^M(D(M))$ where $D(M) := \{P^M(\cdot \mid s,a) \mid s \in \mathcal{S}, a \in \mathcal{A}\}$ is the set of all possible one-step state distributions under the MDP $M$. Since $\mathcal{P}$ has $\sigma_P$-sub-Gaussian noise, we write $s_{t+1} = \bar{p}^M(s_t, a_t) + \epsilon_t^P$ in the natural way. We now use Equation 8 to reduce the regret to a sum of set of widths. To improve notational clarity we write $x_{k,i} = (s_{t_k+i}, a_{t_k+i})$;

$$\mathbb{E}[\Delta_k] \leq \mathbb{E}\Big[\sum_{i=1}^{\tau}\{\bar{r}^k(x_{k,i}) - \bar{r}^*(x_{k,i}) + U_i^k(P^k(x_{k,i})) - U_i^k(P^*(x_{k,i}))\}\Big]$$

$$\leq \mathbb{E}\Big[\sum_{i=1}^{\tau}\{|\bar{r}^k(x_{k,i}) - \bar{r}^*(x_{k,i})| + K^k\|\bar{p}^k(x_{k,i}) - \bar{p}^*(x_{k,i})\|_2\}\Big].$$

To summarize the results until now, we have expressed the regret at time $T$ as the sum of the disagreements between the true model and the estimated model over the full length of the trajectories. Now, using Assumption 3.1, we can bound the RHS of the above equation with some confidence. However, that would grow with the increasing trajectory length and would be a loose bound. Instead, we seek tight bounds and use results from the confidence sets $\mathcal{R}_k := \mathcal{R}_{t_k}(\beta^*(\delta))$ and $\mathcal{P}_k := \mathcal{P}_{t_k}(\beta^*(\delta))$, corresponding to the confidence sets of transition and reward functions respectively.

Let $A = \{R^*, R_k \in \mathcal{R}_k \ \forall k\}$ and $B = \{P^*, P_k \in \mathcal{P}_k \ \forall k\}$. Now, the total regret $\mathbb{E}[\text{Regret}] = \mathbb{E}[\text{Regret } \mathbf{1}_{A \cap B}] + \mathbb{E}[\text{Regret } \mathbf{1}_{A^c \cup B^c}]$. Next we need to compute $\mathbb{P}(A^c \cup B^c)$. We outline the case for $A^c$, $B^c$ would be analogous. The event $A^c$ captures the case that either the true reward function $R^*$ or the sampled reward function $R^k$ do not belong to the confidence sets $\mathcal{R}^k$. This probability by union bound is bounded by $Pr(A^c) \leq 2\delta + 2\delta = 4\delta$. Thus, $Pr(A^c \cup B^c) \leq 8\delta$ [2]. On the failure event $A^c \cup B^c$ we lose the nice "width" control. So we upper bound the worst case regret contribution by constants $\|\bar{r}^k(x_{k,i}) - \bar{r}^*(x_{k,i})\| \leq g_{\mathcal{R}}(t_k)$ and $\|\bar{p}^k(x_{k,i}) - \bar{p}^*(x_{k,i})\| \leq g_{\mathcal{P}}(t_k)$. This gives us the following worst case bound :

Summing over all the episodes, we get:

$$\mathbb{E}[\text{Regret}(T, \pi^{PS}, M^*)] \leq \sum_{k=1}^{m}\sum_{i=1}^{\tau}\{w_{\mathcal{R}_k}(x_{k,i}) + \mathbb{E}[K^k]w_{\mathcal{P}_k}(x_{k,i}) + 8(g_{\mathcal{R}}(t_k) + g_{\mathcal{P}}(t_k))\delta\}$$

Now, using Lemma 5.1 ensures that $\mathbb{E}[K^k] = \mathbb{E}[K^*]$, so that $\mathbb{E}[K^k|A, B] \leq \frac{\mathbb{E}[K^*]}{\mathbb{P}(A,B)} \leq \frac{\mathbb{E}[K^*]}{1-8\delta}$ by a union bound on $\{A^c \cup B^c\}$. Fixing $\delta = 1/8T$ produces:

$$\mathbb{E}[\text{Regret}(T, \pi^{PS}, M^*)] \leq \big(g_{\mathcal{R}}(t_m) + g_{\mathcal{P}}(t_m)\big) + \sum_{k=1}^{m}\sum_{i=1}^{\tau}w_{\mathcal{R}_k}(x_{k,i})$$

$$+ \mathbb{E}[K^*]\Big(1 + \frac{1}{T-1}\Big)\sum_{k=1}^{m}\sum_{i=1}^{\tau}w_{\mathcal{P}_k}(x_{k,i}).$$

---

[2]Here the randomization arises due to the learning process

Next, using the width bounds from Lemma A.2, we get the following:

$$\mathbb{E}[\text{Regret}(T, \pi^{PS}, M^*)] \leq [g_{\mathcal{R}}(t_m) + g_{\mathcal{P}}(t_m)] + R^W + \mathbb{E}[K^*]\left(1 + \frac{1}{T-1}\right)P^W, \tag{6}$$

where

$$R^W := 1 + 2\tau g_{\mathcal{R}}(t_m)\dim_E(\mathcal{R}_m, T^{-1}) + 4\sqrt{\beta^*(1/8T)\dim_E(\mathcal{R}_{t_m}, T^{-1})T}$$

and

$$P^W := 1 + 2\tau g_{\mathcal{P}}(t_m)\dim_E(\mathcal{P}_m, T^{-1}) + 4\sqrt{\beta^*(1/8T)\dim_E(\mathcal{P}_{t_m}, T^{-1})T}.$$

Here, $\dim_E(\mathcal{P}_{t_m}, T^{-1})$ and $\dim_E(\mathcal{R}_m, T^{-1})$ are the eluder dimensions of the transition and reward function classes. Which is a measure of complexity of the function class, discussed in Section 6. $\beta^*(\cdot)$ is a scalar valued function which controls the size of confidence sets, discussed in Section 7.

# 6 FUNCTION CLASS

We begin with the definition of the eluder dimension to quantify the complexity of learning an MDP in an infinite state space setting. Intuitively, the eluder dimension captures the longest possible sequence of inputs $x_1, x_2, x_3, \ldots, x_d$ to a real vector valued function $f$, such that knowing the function values of $f(x_1), f(x_2), \ldots, f(x_i)$ does not reveal the value at $f(x_{i+1})$.

**Definition 6.1** $((\mathcal{F}, \epsilon, \mathcal{X}) - \text{dependence})$. We say that $x \in \mathcal{X}$ is $(\mathcal{F}, \epsilon, \mathcal{X})$-dependent on $\{x_1, x_2, \ldots, x_n\} \subseteq \mathcal{X}$ if $\forall f, \bar{f} \in \mathcal{F}$, $\sum_{i=1}^{n}\left\|f(x_i) - \bar{f}(x_i)\right\|_2^2 \leq \epsilon \implies \|f(x) - \bar{f}(x)\|_2 \leq \epsilon$.
The element $x \in \mathcal{X}$ is $(\epsilon, \mathcal{F})$-independent of $\{x_1, x_2, \ldots, x_n\}$ if it does not satisfy the definition for dependence.

**Definition 6.2** (Eluder Dimension on $\mathcal{X}$). The eluder dimension $\dim_E(\mathcal{F}, \epsilon, \mathcal{X})$ is the length of the longest possible sequence of elements in $\mathcal{X}$ such that, for some $\epsilon' \geq \epsilon$, every element $x \in \mathcal{X}$ is $(\mathcal{F}, \epsilon', \mathcal{X})$-independent of its predecessors.

Classical complexity measures such as the VC dimension are tailored to supervised learning with i.i.d. data and do not capture the intrinsic difficulty of reinforcement learning, where the learner must both explore and control a system while inducing its own data distribution. The eluder dimension (and variants such as the Bellman–eluder dimension) provides an RL-appropriate information measure: it generalizes linear independence to nonlinear function classes by quantifying the longest sequence of points that remain ($\epsilon$-)independent—i.e., not predictable—from prior observations. This measure reflects how many informative interactions are required to learn functions (e.g., rewards, values, or dynamics) to $\epsilon$-accuracy and thereby enable effective control. Throughout, we omit the domain parameter $\mathcal{X}$ when clear from context to simplify notation.

**Definition 6.3** (Set Widths). For any set of functions $\mathcal{F}$ we define the width of the set at $x$ to be the maximum L2 deviation between any two members of $\mathcal{F}$ evaluated at $x$,

$$w_{\mathcal{F}}(x) := \sup_{f^u, f^l \in \mathcal{F}} \|f^u(x) - f^l(x)\|_2$$

**Lemma 6.4** (Bounding the sum of large widths, (Russo & Roy, 2014), Proposition 8). *If* $\{\beta_t > 0 : t \in \mathbb{N}\}$ *is a nondecreasing sequence with* $\mathcal{F}_t = \mathcal{F}_t(\beta_t)$ *then,*

$$\sum_{k=1}^{m}\sum_{i=1}^{\tau}\mathbf{1}\{w_{\mathcal{F}_{t_K}}(x_{t_k+i}) > \epsilon\} \leq \left(\frac{4\beta_T}{\epsilon^2} + \tau\right)dim_E(\mathcal{F}, \epsilon)$$

**Definition 6.5** (Monotonically Increasing Sequence). A sequence $\{C_k : k \in \mathbb{N}\}$ is monotonically increasing if, for every $1 < k < m - 1$, $C_k < C_{k+1}$. This can be interpreted to be the discretized version of the growth function $g_P(t_k) = C_k$.

**Lemma 6.6** (Bounding the sum of widths). *If* $\{\beta_t > 0 : t \in \mathbb{N}\}$ *is a nondecreasing sequence with* $\mathcal{F}_t = \mathcal{F}_t(\beta_t)$ *and* $\|f_t\|_2 \leq C_t$, *for all* $f_t \in \mathcal{F}_t$, *then*

$$\sum_{k=1}^{m}\sum_{i=1}^{\tau}w_{\mathcal{F}_{t_k}}(x_{t_k+i}) \leq 1 + 2\tau C_m\dim_E(\mathcal{F}_{t_m}, T^{-1}) + 4\sqrt{\beta_T\dim_E(\mathcal{F}_{t_m}, T^{-1})T}$$

*Proof* We refer the reader to Appendix A.2 for the proof. □

While it is possible to bound the eluder dimension of a function class for simpler function classes such as generalized linear models (Russo & Roy, 2014), for more complex function classes like deep neural networks this can be fairly large. Rendering the bounds practically invalid. To fix this issue we need a function class which can have the expressibility of a neural network, and at the same time have a provably bounded eluder dimension. Memory-consistent neural networks(MCNN) proposed in Sridhar et al. (2024) are one such class of functions. The eluder dimension of an MCNN function class is bounded by the number of memories, with the choice of $\epsilon$ guided by Lemma 4.6 in (Sridhar et al., 2024). The same applies to transformer models in Sridhar et al. (2025) as well due to similar reasons. This paves the way for interesting research questions to be asked in the future.

## 7 CONFIDENCE SETS

We have established the regret bound in 3 for a possibly unbounded MDP in terms of a growth function and the eluder dimension of the function class, with confidence sets *centered at* least–squares solutions. Assume observations $(x_i, y_i)$ are generated by some $f^* \in \mathcal{F}$. Define the cumulative squared loss

$$L_{2,t}(f) = \sum_{i=1}^{t-1} \|f(x_i) - y_i\|_2^2, \quad \hat{f}_t^{\mathrm{LS}} \in \arg\min_{f \in \mathcal{F}} L_{2,t}(f),$$

and the empirical 2–norm

$$\|g\|_{2,E_t}^2 = \sum_{i=1}^{t-1} \|g(x_i)\|_2^2.$$

The confidence set at time $t$ is the ball

$$\mathcal{F}_t(\beta_t) = \big\{ f \in \mathcal{F} : \|f - \hat{f}_t^{\mathrm{LS}}\|_{2,E_t} \leq \sqrt{\beta_t} \big\},$$

where $\beta_t$ (via the growth function and confidence level) calibrates the radius. Intuitively, $\|f - \hat{f}_t^{\mathrm{LS}}\|_{2,E_t}$ measures the sample-wise *disagreement* between $f$ and the least–squares center. The resulting regret bound scales with the cumulative widths of these sets and is controlled by the eluder dimension.

We are now ready to state the following lemma, which essentially says that the true function $f^*$ is contained in the intersection of a small neighborhood of the least square estimates at different times.

**Lemma 7.1** (Confidence sets with high probability). *For all $\delta > 0$, and the confidence sets $\mathcal{F}_t = \mathcal{F}_t(\beta_t^*(\delta))$, for all $t \in \mathbb{N}$, we have:*

$$\mathbb{P}\left( f^* \in \bigcap_{t=1}^{\infty} \mathcal{F}_t \right) \geq 1 - 2\delta.$$

*Proof* We refer the reader to Appendix A.4 for the proof. □

The consequence of this lemma being it lower bounds the probability that we have the right MDP model M* (reward and transition functions) in a neighborhood of the least squares estimates.

## 8 CONCLUDING REMARKS

Assumption 3.1 provides a tractable starting point for analyzing this setting. A natural extension is to admit more general growth functions $g(t)$ and to develop commensurate analytical tools for such regimes. In addition, it is important to relax environmental passivity by allowing non-passive, potentially adversarial dynamics that adapt to the agent and resist identification and control. The environment studied in the paper can be further generalized by taking explicitly into account the inability to elicit a precise transition distribution $P^{\mathrm{M}}$ over the state-space. In the future, we will extend our findings to the robust MDP case (Bovy et al., 2024; Itoh & Nakamura, 2007), where the attacker is only able to specify portions of the distribution, thus resulting in a credal set, that is, a closed and convex set of probabilities. To do so, we will deploy techniques from the Imprecise

probability literature (Augustin et al., 2014; Walley, 1991; Troffaes & De Cooman, 2014), that allow to derive results in the form of sets and intervals, but which better reflect the intrinsic uncertainty of dealing with partially-known probabilities.

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

## A  APPENDIX

### A.1  EPISODIC REGRET ANALYSIS

**Lemma A.1.** *The episodic regret under dynamic programming according to definition 5.2 is given by* $\mathbb{E}[\Delta_k] = \sum_{i=1}^{\tau} \mathbb{E}\left[\left((T_{\mu_k}^{M_k} - T_{\mu_k}^{M^*}) V_{\mu_k,i+1}^{M_k}\right)(s_{t_k+i})\right]$

*Proof* For brevity, write
$$T_k^* := T_{\mu_k}^{M^*}, \quad T_k^k := T_{\mu_k}^{M_k}, \quad V_{k,i}^* := V_{\mu_k,i}^{M^*}, \quad V_{k,i}^k := V_{\mu_k,i}^{M_k},$$
and focus on the integrand in the regret expression in isolation. For an episode which starts at state $s_0$, we define the episode-$k$ regret
$$\mathcal{D}_k := V_{\mu^*,1}^{M^*}(s_0) - V_{\mu_k,1}^{M^*}(s_0) = V_{*,1}^*(s_0) - V_{k,1}^*(s_0).$$

Add and subtract $V_{k,1}^k(s_0)$:
$$\mathcal{D}_k = \underbrace{V_{*,1}^*(s_0) - V_{k,1}^k(s_0)}_{(A)} + \underbrace{V_{k,1}^k(s_0) - V_{k,1}^*(s_0)}_{(B)}. \tag{7}$$

Taking $g(M) = V_{\mu^*,1}^M(s_0) - V_{\mu_k,1}^M(s_0)$ in Lemma 5.1 and noting $\mu_k$ is $H_{t_k}$-measurable,
$$\mathbb{E}[(A) \mid H_{t_k}] = 0 \quad \Rightarrow \quad \mathbb{E}[\mathcal{D}_k \mid H_{t_k}] = \mathbb{E}\left[V_{k,1}^k(s_0) - V_{k,1}^*(s_0) \mid H_{t_k}\right].$$
Hence, it suffices to analyze the model gap for the fixed policy $\mu_k$. Let the (random) states within episode $k$ under the *true* dynamics and policy $\mu_k$ be $s_{t_k+1}, \ldots, s_{t_k+\tau}$. Define for $i = 1, \ldots, \tau$,
$$\delta_i := V_{k,i}^k(s_{t_k+i}) - V_{k,i}^*(s_{t_k+i}), \qquad \delta_{\tau+1} := 0.$$
Using $V_{k,i}^k = T_k^k V_{k,i+1}^k$ and $V_{k,i}^* = T_k^* V_{k,i+1}^*$,
$$\delta_i = \left(T_k^k V_{k,i+1}^k\right)(s_{t_k+i}) - \left(T_k^* V_{k,i+1}^*\right)(s_{t_k+i})$$
$$= \underbrace{\left((T_k^k - T_k^*) V_{k,i+1}^k\right)(s_{t_k+i})}_{=:b_i} + \left(T_k^*(V_{k,i+1}^k - V_{k,i+1}^*)\right)(s_{t_k+i})$$
$$= b_i + \mathbb{E}[\delta_{i+1} \mid H_{t_k+i}],$$
where the last equality uses the true transition kernel inside $T_k$. Note that in the above expression $b_i$ is the Bellman error, and $\left(T_k^*(V_{k,i+1}^k - V_{k,i+1}^*)\right)(s_{t_k+i})$ is the next step value; it is the expected $\delta_{i+1}$ given the current state, because $T_k^*$ uses the true transition probabilities. $\left(T_k^*(V_{k,i+1}^k - V_{k,i+1}^*)\right)(s_{t_k+i}) = \mathbb{E}[\delta_{i+1} \mid s_{t_k+i}]$ We expand the above expression again to observe the telescoping pattern
$$\delta_1 = b_1 + \mathbb{E}[\delta_2 \mid H_{t_k+1}]$$
and
$$\delta_2 = b_2 + \mathbb{E}[\delta_3 \mid H_{t_k+2}]$$
and so on. Therefore,
$$\mathbb{E}[\delta_1] = \mathbb{E}[b_1] + \mathbb{E}[\mathbb{E}[\delta_2 \mid H_{t_k+1}]] = \mathbb{E}[b_1] + \mathbb{E}[\delta_2].$$
Taking expectations of $\delta_i$ and summing $i = 1, \ldots, \tau$, the $\mathbb{E}[\delta_{i+1}]$ terms cancel, except the last one that has zero mean (with $\delta_{\tau+1} = 0$). Therefore,
$$\mathbb{E}[\delta_1] = \sum_{i=1}^{\tau} \mathbb{E}[b_i].$$

Since $\delta_1 = V_{k,1}^k(s_0) - V_{k,1}^*(s_0)$, combining with (7) gives
$$\mathbb{E}[\mathcal{D}_k] = \mathbb{E}\left[V_{k,1}^k(s_0) - V_{k,1}^*(s_0)\right] = \sum_{i=1}^{\tau} \mathbb{E}\left[\left((T_k^k - T_k^*)V_{k,i+1}^k\right)(s_{t_k+i})\right].$$

Now, remember that
$$\Delta_k := \int_{s_0 \in \mathcal{S}} \mathcal{D}_k \rho(s_0) \mathrm{d}s_0 = \mathbb{E}[\mathcal{D}_k].$$
Therefore, taking the expectation, we get,
$$\mathbb{E}[\Delta_k] = \sum_{i=1}^{\tau} \mathbb{E}\left[\left((T_{\mu_k}^{M_k} - T_{\mu_k}^{M^*}) V_{\mu_k,i+1}^{M_k}\right)(s_{t_k+i})\right], \tag{8}$$

## A.2 SUM OF WIDTHS BOUND

**Lemma A.2** (Bounding the sum of widths). *If $\{\beta_t > 0 : t \in \mathbb{N}\}$ is a nondecreasing sequence with $\mathcal{F}_t = \mathcal{F}_t(\beta_t)$ and $\|f_t\|_2 \leq C_t$, for all $f_t \in \mathcal{F}_t$, then*

$$\sum_{k=1}^{m}\sum_{i=1}^{\tau} w_{\mathcal{F}_{t_k}}(x_{t_k+i}) \leq 1 + 2\tau C_m \dim_E(\mathcal{F}_{t_m}, T^{-1}) + 4\sqrt{\beta_T \dim_E(\mathcal{F}_{t_m}, T^{-1})T}$$

*Proof* At any time $i$, $w_{\mathcal{F}_{t_k}}(x_{t_k+i}) \leq 2C_m$. This is because at each time $t_k$, $\|f_k\|_2 \leq C_k$, meaning $w_{\mathcal{F}_{t_k}}(x_{t_k+i}) \leq 2C_k \leq 2C_m$, since the sequence $C_k$ is monotonically increasing in $k$. For ease of notation, we denote $d = \dim_E(\mathcal{F}_{t_m}, T^{-1})$ and $w_t = w_{\mathcal{F}_{t_k}}(x_{t_k+i})$. We use the Eluder dimension of the last function class as the absolute upper bound, since $\dim_E(\mathcal{F}_{t_1}, T^{-1}) \leq \dim_E(\mathcal{F}_{t_2}, T^{-1}) \leq \cdots \leq \dim_E(\mathcal{F}_{t_m}, T^{-1})$.

The first step is to reorder the sequence $(w_1, \ldots, w_T) \to (w_{i_1}, \ldots w_{i_T})$ where $w_{i_1} \geq w_{i_2} \geq \cdots \geq w_{i_T}$. This lets us write a more civilized version of the summation as the following:

$$\sum_{k=1}^{m}\sum_{i=1}^{\tau} w_{\mathcal{F}_{t_K}}(x_{t_k+i}) = \sum_{t=1}^{T} w_{i_t} = \sum_{t=1}^{T} w_{i_t}\mathbf{1}\{w_{i_t} \leq T^{-1}\} + \sum_{t=1}^{T} w_{i_t}\mathbf{1}\{w_{i_t} > T^{-1}\}$$

$$\leq 1 + \sum_{t=1}^{T} w_{i_t}\mathbf{1}\{w_{i_t} > T^{-1}\}.$$

Now, $w_{i_t} \leq 2C_m$. Also, remember that $w_{i_t} > \epsilon \iff \sum_{t=1}^{T}\mathbf{1}\{w_{i_t} > \epsilon\} \geq t$, meaning $t \leq (\frac{4\beta_T}{\epsilon^2} + \tau)d$. Therefore, $\epsilon < \sqrt{\frac{4\beta_T d}{t - \tau d}}$, from Lemma 6.4. Thus, whenever, $w_{i_t} > \epsilon \geq T^{-1}$, then $\epsilon$ is strictly less than $\sqrt{\frac{4\beta_T d}{t - \tau d}}$. Thus, if $w_{i_t} > T^{-1}$ then it is upper bounded by the minimum of the two upper bounds. That is, $w_{i_t} \leq \min\{2C_m, \sqrt{\frac{4\beta_T d}{t - \tau d}}\}$. Therefore,

$$\sum_{t=1}^{T} w_{i_t}\mathbf{1}\{w_{i_t} > T^{-1}\} \leq 2\tau C_m d + \sum_{t=\tau d+1}^{T} \sqrt{\frac{4\beta_T d}{t - \tau d}}$$

$$\leq 2\tau C_m d + 2\sqrt{\beta_T}\int_0^T \sqrt{\frac{d}{t}}\mathrm{d}t \leq 2\tau C_m d + 4\sqrt{\beta_T d T}.$$

The first inequality comes from the fact that $w_{i_t}$ is monotonically decreasing, and $\sqrt{\frac{4\beta_T d}{t - \tau d}}$ tends to $\infty$ when $t = \tau d$, meaning that for $t < \tau d$, $w_{i_t}$ is upper bounded by $2C_m$. $\qquad\square$

## A.3 CONCENTRATION GUARANTEE

**Lemma A.3.** *(Exponential Martingale) Assume a real-valued random variable $Z_i$ adapted to $\mathcal{H}_i$. The conditional mean is $\mu_i = \mathbb{E}[Z_i|\mathcal{H}_{i-1}]$ and the conditional cumulant generating function is $\psi_i(\lambda) = \log\mathbb{E}[\exp(\lambda(Z_i - \mu_i))|\mathcal{H}_{i-1}]$. Then,*

$$M_n(\lambda) = \exp\left(\sum_1^n \lambda(Z_i - \mu_i) - \psi_i(\lambda)\right)$$

*is a martingale with $\mathbb{E}[M_n(\lambda)] = 1$.*

**Lemma A.4.** *(Concentration Guarantee) Let $Z_i$ be a real random variable adapted to $\mathcal{H}_i$. We define the conditional mean $\mu_i = \mathbb{E}[Z_i|\mathcal{H}_{i-1}]$ and conditional cumulant generating function $\psi_i(\lambda) = \log\mathbb{E}[\exp(\lambda(Z_i - \mu_i))|\mathcal{H}_{i-1}]$. Then,*

$$\mathbb{P}\left(\bigcup_{n=1}^{\infty}\{\sum_{1}^{n}\lambda(Z_i - \mu_i) - \psi_i(\lambda) \geq x\}\right) < e^{-x}.$$

*Proof* From Lemma A.3, we know that $M_n(\lambda)$ is a martingale, and that the expected value is $\mathbb{E}[M_n(\lambda)] = 1$. Now, consider the event

$$A = \bigcup_{k=1}^{n}\{M_k(\lambda) \geq x\},$$

and an event

$$B = \{M_{\tau_x}(\lambda) \geq x\},$$

where $\tau_x = \inf\{n \geq 0 : M_n(\lambda) \geq x\}$.

Note that $A$ and $B$ are the same event. It is easier to reason using the complement of these events. Event $A^c$ is true when none of the values of $M_k(\lambda) \geq x$, which is same as saying there does not exist a *crossing* time $\tau_x$ for which $M_n(\lambda) \geq x$, which is event $B^c$. Therefore $A$ and $B$ are the same event.

Therefore, using Markov's inequality:

$$x\mathbb{P}(M_{\tau_x}) \leq \mathbb{E}M_{\tau_x} = 1,$$

meaning,

$$\mathbb{P}\left(\bigcup_{k=1}^{n}\{M_k(\lambda) \geq x\}\right) \leq \frac{1}{x}.$$

Since, for any $n \geq 1$ and $x \geq 0$ the above statement is true, then using the monotone convergence theorem:

$$\mathbb{P}\left(\bigcup_{k=1}^{\infty}\{M_k(\lambda) \geq x\}\right) \leq \frac{1}{x},$$

or

$$\mathbb{P}\left(\bigcup_{k=1}^{\infty}\{M_k(\lambda) \geq e^x\}\right) \leq e^{-x}.$$

Since exponentials are strictly monotonic functions, and using the definition of $M_n(\lambda)$ from Lemma A.3, we get the following:

$$\mathbb{P}\left(\bigcup_{n=1}^{\infty}\{\sum_{1}^{n}\lambda(Z_i - \mu_i) - \psi_i(\lambda) \geq x\}\right) < e^{-x}.$$

$\square$

### A.4 High Probability Confidence Sets

**Lemma A.5** (Confidence sets with high probability). *For all $\delta > 0$, and the confidence sets $\mathcal{F}_t = \mathcal{F}_t(\beta_t^*(\delta))$, for all $t \in \mathbb{N}$, we have:*

$$\mathbb{P}\left(f^* \in \bigcap_{t=1}^{\infty}\mathcal{F}_t\right) \geq 1 - 2\delta.$$

*Proof* Consider an arbitrary $f \in \mathcal{F}$. We use the notational shorthand $f_t^* = f^*(x_t)$ and $f_t = f(x_t)$ for $f, f^*$ evaluated at an arbitrary point $x_t$. The function maps values to the vector space $\mathcal{Y} \subset \mathbb{R}^d$ where the inner product $<y, y> = \|y\|_2^2$. We define

$$\begin{aligned}
Z_t &= \|f_t^* - y_t\|_2^2 - \|f_t - y_t\|_2^2 \\
&= \|f_t^* - y_t\|_2^2 - \|f_t - f_t^* + f_t^* - y_t\|_2^2 \\
&= \|f_t^* - y_t\|_2^2 - \|f_t - f_t^*\|_2^2 - \|f_t^* - y_t\|_2 + 2 < f_t - f_t^*, f_t^* - y_t > \\
&= -\|f_t - f_t^*\|_2^2 + 2 < f_t - f_t^*, \epsilon_t >,
\end{aligned}$$

where $\epsilon_t = f_t^* - y_t$. So, clearly, $\mu_t = -\|f_t - f_t^*\|_2$. Since the noise is $\sigma$-sub-Gaussian, then

$$\mathbb{E}[\exp(<\phi, \epsilon>)] \leq \exp\left(\frac{\|\phi\|_2^2 \sigma^2}{2}\right), \quad \forall \phi \in \mathcal{Y}.$$

Using this:

$$\begin{aligned}
\psi_t(\lambda) &= \log \mathbb{E}\big[\exp(\lambda(Z_t - \mu_t))|\mathcal{H}_{t-1}\big] \\
&= \log \mathbb{E}[\exp(2\lambda < f_t - f_t^*, \epsilon_t >)] \\
&\leq \frac{\|2\lambda(f_t - f_t^*)\|_2^2 \sigma^2}{2}.
\end{aligned}$$

The point of assuming that the noise is sub-Gaussian is to show that the noise is upper bounded by a scaled normal distribution. Intuitively, $Z_t$ can be interpreted as the difference of errors between $f$ and $f^*$ with respect to the observed value $y_t$.

We apply Lemma A.4 with $\lambda = \frac{1}{4\sigma^2}$ and $x = log(1/\delta)$ to get:

$$\mathbb{P}\left(\bigcup_{t=1}^{\infty}\left\{\left(\sum_{1}^{t}\frac{1}{4\sigma^2}(Z_i - \mu_i) - \psi_i\left(\frac{1}{4\sigma^2}\right)\right) \geq \log(1/\delta)\right\}\right) \leq e^{\log \delta}$$

$$\mathbb{P}\left(\bigcap_{t=1}^{\infty}\left\{\left(\sum_{1}^{t}\frac{1}{4\sigma^2}(Z_i - \mu_i) - \psi_i\left(\frac{1}{4\sigma^2}\right)\right) \leq \log(1/\delta)\right\}\right) \geq 1 - \delta$$

$$\mathbb{P}\left(\bigcap_{t=1}^{\infty}\left\{\left(\sum_{1}^{t}(Z_i - \mu_i) - 4\sigma^2 \psi_i\left(\frac{1}{4\sigma^2}\right)\right) \leq 4\sigma^2 \log(1/\delta)\right\}\right) \geq 1 - \delta$$

$$\mathbb{P}\left(\bigcap_{t=1}^{\infty}\left\{\left(\sum_{1}^{t}(Z_i - \mu_i) - 4\sigma^2 \psi_i\left(\frac{1}{4\sigma^2}\right)\right) - 4\sigma^2 \log(1/\delta) \leq 0\right\}\right) \geq 1 - \delta$$

$$\mathbb{P}\left(\bigcap_{t=1}^{\infty}\left\{\left(\sum_{1}^{t}(Z_i - \mu_i) - \frac{\|f_i - f_i^*\|_2^2}{2}\right) - 4\sigma^2 \log(1/\delta) \leq 0\right\}\right) \geq 1 - \delta$$

$$\mathbb{P}\left(\bigcap_{t=1}^{\infty}\left\{\left(\sum_{1}^{t}Z_i + \frac{\|f_i - f_i^*\|_2^2}{2}\right) - 4\sigma^2 \log(1/\delta) \leq 0\right\}\right) \geq 1 - \delta,$$

over a length of time this is essentially the difference in loss function, $\sum_{i=1}^{t-1}Z_i = L_{2,t}(f^*) - L_{2,t}(f)$. Since the above is true $\forall t$, we have:

$$\mathbb{P}\left(\left\{L_{2,t}(f^*) - L_{2,t}(f) + \frac{1}{2}\|f - f^*\|_{2,E_t}^2 - 4\sigma^2 \log(1/\delta) \leq 0, \forall t\right\}\right) \geq 1 - \delta.$$

Rearranging the above terms,

$$\mathbb{P}\left(\left\{L_{2,t}(f) \geq L_{2,t}(f^*) + \frac{1}{2}\|f - f^*\|_{2,E_t}^2 - 4\sigma^2 \log(1/\delta), \forall t\right\}\right) \geq 1 - \delta.$$

The above is true for any function $f \in \mathcal{F}$ around $f^*$ with probability at least $1 - \delta$.

Now, consider the data dependent minimizer $\hat{f}_t \in \arg\min_{f \in \mathcal{F}} L_{2,t}(f)$, for which, by optimality of $\hat{f}_t$, $L_{2,t}(\hat{f}_t) \leq L_{2,t}(f^*)$. That is, $0 \geq L_{2,t}(\hat{f}_t) - L_{2,t}(f^*)$. In other words,

$$\frac{1}{2}\|\hat{f}_t - f^*\|_{2,E_t}^2 - 4\sigma^2 \log(1/\delta) \geq 0$$

Thus, both $\hat{f}_t$ and $f^*$ can violate this with probability at most $\delta$. Next, using the union bound with probability at least $1 - 2\delta$, the following is true $\forall t$,

$$\|\hat{f}_t - f^*\|_{2,E_t} \leq \sqrt{8\sigma^2 \log(1/\delta)} = \sqrt{\beta^*(\delta)}.$$

The the true function $f^*$ is contained in a neighborhood of the least square minimizer $\hat{f}_t$, $\forall t$ with probability at least $1 - 2\delta$.

