# OpenReview forum: "Hunting Games"
_ICLR.cc/2026/Conference — Submitted to ICLR 2026_

### Official Review · Reviewer_kCXi · 2025-10-27

**Soundness:** 3
**Presentation:** 3
**Contribution:** 2
**Rating:** 4
**Confidence:** 2

**Summary:**

This paper introduces and formalizes a novel problem setting in reinforcement learning (RL) termed the "hunting game." This setting captures environments where an agent is continually pushed into previously unseen regions of the state space, not due to an adversarial opponent, but due to the environment's own "heavy-tailed" dynamics and a temporal misalignment between the agent's decision epochs and the environment's natural evolution. It gives a new formalism (the hunting game) and a corresponding family of growth-controlled MDPs, with regret analysis for Posterior Sampling RL (PSRL) in this setting.

**Strengths:**

1. The Growth-Weighted Eluder Scaling is an original theoretical advancement, creatively combining established concepts. This paper uniquely shows how temporal environmental growth (g(t)) acts as a multiplier on dimE. This unified framework is highly novel in analyzing the trade-off between structural complexity and non-stationarity.
2. The paper excels in its clarity by providing a structured roadmap of its theoretical argument. The relationship between the key components—the growth functions g(t), the model complexity dimE , and the eventual regret Regret(T)—is clearly articulated in the discussion sections (Section 4 and 8).
3. Crucially, the paper offers a clear, actionable synthesis of its findings. It concludes that for the Hunting Game to be worthwhile (i.e., achieving sublinear regret), two necessary conditions must be satisfied. This clear translation from theory to operational constraints (e.g., controlling decision-epoch frequency) strengthens the paper's overall message and readability.

**Weaknesses:**

1. Sub-Gaussian Noise versus Heavy-Tailed MotivationThe paper suffers from a conceptual contradiction between its motivation and its formal assumption. The core problem is motivated by environments exhibiting "heavy-tailed variability" and high-impact surprises. In probability theory, heavy tails imply high risk and often unbounded variance. However, Assumption 3.1 mandates that all noise processes are σ-sub-Gaussian. Sub-Gaussianity implies light tails and is incompatible with the genuine heavy-tailed stochasticity cited in the motivation.
2. Insufficient Quantification of Required Growth Rate g(t). The paper correctly asserts that the growth function g(t) must be "slowly growing" for sublinear regret. However, this qualitative statement lacks the crucial quantitative constraint necessary to fulfill the paper's promise of providing actionable operational levers.
3. Missing Algorithmic Implementation.The work is entirely theoretical and lacks an empirical section. A critical evaluation of PSRL in this new formalism requires demonstrating the practical feasibility of two key steps: Model Sampling: How efficiently can Mk be sampled from the posterior when the underlying model classes (R, P) are complex, continuous function approximators (like neural networks) defined over a continuously expanding state support Sk? Policy Computation: How is the optimal policy μk = μMk computed for the growing subset Sk during episode k(Algorithm 1)?

**Questions:**

1. Empirical Demonstration: Could the authors provide a simple, illustrative experiment? For instance, a synthetic MDP where the state space expands along one dimension according to a known gP(t).
2. Tightness of the Bound: Is the derived regret bound (Eq. 6) tight? Can the authors provide, or point to the possibility of, a matching lower bound that also depends on the product of the eluder dimension and the growth function?
3. The analysis suggests that "bounding novelty via the growth functions" is a path to sublinear regret. How could an agent or a system designer operationally achieve this?
4. How critical is the sub-Gaussian noise assumption? Could the analysis be extended to handle truly heavy-tailed noise, which might be a more accurate model for the "impactful surprises" mentioned in the abstract?

---

> ### Author Response · Authors · 2025-11-26
> **Response to Reviewer kCXi**
>
> We thank the reviewer for the strengths pointed out in the paper and appreciate the feedback. Please let us know if you have lingering questions and whether we can provide any additional clarifications during the discussion period.
>
> * Sub-Gaussian Noise versus Heavy-Tailed Motivation
>
> In this paper we do not solve the ”heavy tailed” aspect of the distribution. Instead we assume that the growth of the mean of observed states (and rewards) is bounded by functions $g_P$ (and $g_R$) respectively.
>
> *  Insufficient Quantification of Required Growth Rate g(t)
>
> The quantification of the regret in Equation-3, inherently bounds the regret in terms of the growth rate and other parameters of the PSRL algorithm, such as Eluder dimension, episode length and alike.
>
> * Missing Algorithmic Implementation… How efficiently can Mk be sampled from the posterior …. ? How is the optimal policy μk = μMk computed  … ?
>
> Our contribution is intentionally theoretical: it characterizes the fundamental obstacles and structural conditions governing learning in hunting games. These insights serve as guidance for future algorithmic development, which we view as a promising direction for subsequent work.
>
> We address the questions next.
>
> * Empirical Demonstration: Could the authors provide a simple, illustrative experiment? For instance, a synthetic MDP where the state space expands along one dimension according to a known $g_P(t)$.
>
> We are able to provide this; however, preparing it would require more time than the discussion window permits. We prioritized giving a timely response to the remaining concerns.
>
> * Tightness of the Bound: Is the derived regret bound (Eq. 6) tight? Can the authors provide, or point to the possibility of, a matching lower bound that also depends on the product of the eluder dimension and the growth function?
>
> The bound presented in Equation 3 is a high confidence bound. It is tight when the estimated MDP M_k matches the true model M* . Also, a corollary to Equation 3 is, when the growth bounds are fixed quantities, that is $ g_P = c_1 $ and $ g_R = c_2 $ for all times. In that case this is precisely the bounded state regret bounds computed in (Russo and Roy 2014).
>
> * … "bounding novelty via the growth functions" .. How could an agent or a system designer operationally achieve this?
>
> In short, yes. This bound is precisely stated in Assumption 3.1. We believe that it is not practically unrealistic to expect that physical systems introduce only a limited quantity of novel behavior in each time step, and by extension in each episode.
>
> * How critical is the sub-Gaussian noise assumption?
>
> This assumption is crucial to achieve the high-confidence bound in Lemma 7.1.

---

### Official Review · Reviewer_WJSU · 2025-11-01

**Soundness:** 1
**Presentation:** 1
**Contribution:** 1
**Rating:** 0
**Confidence:** 3

**Summary:**

The paper proposes a new setting named "Hunting Games." Please see the weaknesses section.

**Strengths:**

--

**Weaknesses:**

I suspect a substantial portion of this paper might have been generated by a language model.

My main concern about the paper is clarity. The text is hard-to-follow; most of the time I found myself trying to predict what the authors could have meant. For example, the first sentence of the paper is as follows: 'A plethora of real-world control problems entail an agent learning while the “operational envelope” of the environment is being revealed over time.' Without further knowledge, what operational envelope means is quite unclear. The text continues with "For example, in cyber defense and counter-drone operations, coarse and event-driven interventions are interleaved with long intervals of unobserved evolution (National Security Agency (NSA), 2019; Mandiant Consulting, 2025; Seidaliyeva et al., 2023; Director, Operational Test & Evaluation (DOT&E), 2020). In such cases, the next observed state can jump “far” from where a conventional model expects it to be." This is only an example and I think such statements exist throughout the paper. The paper's language overcomplicates the problem.

I believe the introduction could introduce the problem at hand better. Since the paper proposes a new formalism, a full concrete example that showcases the main aspects of the setting would provide the reader a better grasp of what the problem is and how the existing framework does not capture the main facets. One can, of course, convey the problem differently but I do not think that the paper's introduction does a good job.

Section 3 (Problem Formulation) starts with the following description: "We begin with the set up of the learning problem as an interactive exercise between a hunting agent A and a dynamic target represented by the environment M. It proceeds as repeated interactions over time. At each time instant, the hunting agent A takes an action on M, and receives a scalar reward. This action pushes the target M to react, by changing its state according to some unknown but stationary transition distribution. The hunter agent does not have prior knowledge of the reward function, or the internal model which dictates the evolution of M ’s state. However, through repeated engagements in this interactive process, it improves its understanding of the transition function of M and maximizes the collected reward." This explanation sounds like the standard RL problem, and the 'hunting' reference seems unnecessary. The description between lines 180-188 is also quite cryptic.

There also appears to be several issues with Section 3. First, some sentences are quite similar to the ones paper [1].
Some sentences from this paper:
- The reinforcement learning (RL) agent A interacts in an episodic fashion at times $tk = (k − 1)τ + 1, k = 1, 2, ...$ over time.
- We denote a finite history $H_t = (s_1, a_1, r_1, . . . , s_{t−1}, a_{t−1}, r_{t−1})$ as the sequence of observations made prior to time t.
- The regret incurred by the RL algorithm $\pi$ up to time $T$ is expressed as the following: ...

Some sentences from [1]:
- The reinforcement learning agent interacts with the MDP over episodes that begin at times $t_k = (k − 1)τ + 1, k = 1, 2, . . .$.
- Let $H_t = (s_1, a_1, r_1, ... , s_{t−1}, a_{t−1}, r_{t−1})$ denote the history of observations made prior to time t.
- We define the regret incurred by a reinforcement learning algorithm $\pi$ up to time $T$ to be ...

Please note that these are only some examples and one can find more of these cases. While these resemblances might be okay in separate cases, I believe the structure and the style is also quite similar. The notations and many other definitions are also quite close to [1]. Lemma 5.1 in this paper and Lemma 1 in [1] are exactly the same with $f$ replaced by $\phi$. However, there is no references given. Definitions 5.2 in this paper and Lemma 2 in [1] are again the same.

There are also problems with the logical ordering of the paper. The simplest example is that the result in Section 4 is in terms of Eluder dimension, whose definition is given in Section 6.

There are many more issues and, at any point, I feel like I am trying to decrypt instead of read the paper. This looks very much like an LLM submission.

Minor issues:
- In line 104, the abbreviation BE for Bellman Eluder is used without introducing it first.

[1] Osband, I., Russo, D. and Van Roy, B., 2013. (More) efficient reinforcement learning via posterior sampling. _Advances in Neural Information Processing Systems_, _26_.

**Questions:**

--

**Details Of Ethics Concerns:**

Partially explained in the weaknesses section. The similarities between this paper and [1] are of concern. Some statements are directly taken from [1], a deeper look might reveal more resemblances to more papers.

[1] Osband, I., Russo, D. and Van Roy, B., 2013. (More) efficient reinforcement learning via posterior sampling. Advances in Neural Information Processing Systems, 26.

---

> ### Author Response · Authors · 2025-11-26
> **Response to Reviewer WJSU (part 1/2)**
>
> * I suspect a substantial portion of this paper might have been generated by a language model.
>
> We are sorry that the reviewer had this impression. We can fully guarantee that the paper is humanly written.
>
> * My main concern about the paper is clarity. The text is hard-to-follow; most of the time I found myself trying to predict what the authors could have meant. For example, the first sentence of the paper is as follows: 'A plethora of real-world control problems entail an agent learning while the “operational envelope” of the environment is being revealed over time.' Without further knowledge, what operational envelope means is quite unclear. The text continues with "For example, in cyber defense and counter-drone operations, coarse and event-driven interventions are interleaved with long intervals of unobserved evolution (National Security Agency (NSA), 2019; Mandiant Consulting, 2025; Seidaliyeva et al., 2023; Director, Operational Test & Evaluation (DOT&E), 2020). In such cases, the next observed state can jump “far” from where a conventional model expects it to be." This is only an example and I think such statements exist throughout the paper. The paper's language overcomplicates the problem.
>
>
> While we understand that our writing in some parts of the manuscript may seem hard to parse, we are rather surprised that this is such a concern that warranted a score of 0. That being said, we will improve our writing, so that it does not seem to the reader that our language overcomplicates the problem. For example, in the two reported excerpts, we will clarify the meaning of “operational envelope”, and make it clearer that “In cyber defence and counter-drone work, the user mostly steps in only when something happens, and those actions are spaced out by long periods where they cannot see what’s going on.”
>
> * Section 3 (Problem Formulation) starts with the following description: "We begin with the set up of the learning problem as an interactive exercise between a hunting agent A and a dynamic target represented by the environment M. It proceeds as repeated interactions over time. At each time instant, the hunting agent A takes an action on M, and receives a scalar reward. This action pushes the target M to react, by changing its state according to some unknown but stationary transition distribution. The hunter agent does not have prior knowledge of the reward function, or the internal model which dictates the evolution of M ’s state. However, through repeated engagements in this interactive process, it improves its understanding of the transition function of M and maximizes the collected reward." This explanation sounds like the standard RL problem, and the 'hunting' reference seems unnecessary. The description between lines 180-188 is also quite cryptic.
>
> We agree with the reviewer: we framed our hunting game in a way that it does not depart heavily from the traditional reinforcement learning problems. What is new is the interpretation of the elements at play, and the results that we find. In the updated version, we also make it clearer what we mean by the description in lines 180-188; we write the following.
>
> "In the situation described above, the target M is much weaker than the hunter. It can’t directly harm the latter, and mainly just wants to survive; The target has an advantage because the hunter does not fully understand how the target behaves or changes over time. Even though the hunter can attack, it faces two challenges:
>
> * It doesn’t know how efficient each attack will be in different situations (the reward is unknown).
> * It doesn’t know how its attacks will make the target change from one state to another (the transition probabilities are unknown).
>
> Because of this, the hunter has to learn through trial and error. Over time, as it learns, it improves at attacking the target and moves closer to achieving its goal."
>
> *Some sentences are quite similar to the ones paper [1]. I believe the structure and the style is also quite similar. The notations and many other definitions are also quite close to [1]. Lemma 5.1 in this paper and Lemma 1 in [1]. However, there is no references given. Definitions 5.2 in this paper and Lemma 2 in [1] are again the same.
>
> The reviewer is right; as [1] is a seminal work in RL, we borrow – citing and never stealing – some notation and terminology for our paper. In the updated version of our manuscript, we will explicitly acknowledge when our results (that were presented without proof precisely because follow form [1]) are mutuated from another paper, and where to find them in the original reference.

---

> ### Author Response · Authors · 2025-11-26
> **Response to Reviewer WJSU (part 2/2)**
>
> * There are also problems with the logical ordering of the paper. The simplest example is that the result in Section 4 is in terms of Eluder dimension, whose definition is given in Section 6.
>
> While this is true, and in the updated version of the manuscript we will bring forth the definition of the eluder dimension in our context, it is safe to say that the eluder dimension (ED) is a well-known concept in classic RL, that is why we allowed ourselves to define it properly for our case only later. This also connects with the reviewer’s previous comment, pointing out that our framework resembles that of classic RL. Precisely because of that similitude, we postponed the definition of ED in the hunting game scenario.
>
> * There are many more issues and, at any point, I feel like I am trying to decrypt instead of read the paper. This looks very much like an LLM submission.
>
> Once again, we are sorry that the reviewer feels this way, but ours is a 100% human submission. If they point out other flaws explicitly, we are happy to address them in the updated version of the manuscript.
>
> * In line 104, the abbreviation BE for Bellman Eluder is used without introducing it first.
>
> We agree with the reviewer and we have updated this in our manuscript.

---

### Official Review · Reviewer_YhQ4 · 2025-11-01

**Soundness:** 3
**Presentation:** 3
**Contribution:** 3
**Rating:** 8
**Confidence:** 2

**Summary:**

This paper studies the learning difficulty in MDPs where the ranges of expected rewards and next states grow along with time. The paper introduces a new concept of "hunting game" with bounded growth functions to formalize such environments (target), which leads to forced exploration of the learning agents (hunter). A sublinear regret is proved under the Posterior Sampling for Reinforcement Learning (PSRL) algorithm, which allows the learning agent to interact in an episodic fashion. The regret bound contains width quantities in the form of the eluder dimension multiplied by the growth factor, which implies that both exploration growth and model class complexity lead to the difficulty of learning.

**Strengths:**

1. The observation that environments with growing requirements of exploration can slow down learning even without explicit adversarial behaviors is novel.

2. Different from classical regret bounds that control uncertainty uniformly over the state-action space, this work decomposes regret into set widths along true trajectories.

3. The "hunting game" definition eventually induces a growth-function-related sublinear regret with high confidence, which is intuitively sound.

4. The paper connects the concepts of structure-aware exploration and open-world learning, which is insightful for reinforcement learning in MDPs.

5. The paper is very well-written and easy to follow. The authors decompose the major proof in an elegant manner.

**Weaknesses:**

As I am not familiar with the related works in this field, I did not go through the paper's math. That said, I do not find any factual errors myself and will look to other reviewers who show greater expertise.

**Questions:**

Where do you use the notation $\mathcal{S}_k$ defined in line 194? If I understand correctly, it is different from the $S_k$ in line 225?

---

> ### Author Response · Authors · 2025-11-26
> **Response to Reviewer YhQ4**
>
> We thank the reviewer for the insightful feedback and the positive encouragement on this paper. Please let us know if you have lingering questions and whether we can provide any additional clarifications during the discussion period.
>
> * Where do you use the notation  $S_k$  defined in line 194? If I understand correctly, it is different from the $S_k$  in line 225?
>
> It is indeed the same subset $S_k$ in both lines 194 and 225. It used to define the set of states over which regret is defined at every episode as outlined in Equation 2.

---

### Official Review · Reviewer_gYu8 · 2025-11-01

**Soundness:** 1
**Presentation:** 1
**Contribution:** 1
**Rating:** 2
**Confidence:** 3

**Summary:**

This paper introduces "hunting games". However, there is no clear problem formulation. I had difficulty reading it as explained in the weaknesses in detail.

**Strengths:**

The paper claims to address the practical issue of changes in the operational limits in practice.

**Weaknesses:**

- The problem formulation is not clear. It starts with a standard MDP formulation but some "bounded growth" is assumed in Assumption 3.1. This would have implied time-varying MDP yet there is no discussion on that.
- Reward function is defined as R:S\times A \rigtharrow (0,1] at line 163. However, at line 173, we have r\sim R(s,a) as if it is a probability distribution.
- At line 195, \mathcal{P}_{C,\sigma}^{X,Y} is introduced as the family of distributions from X to Y with l2-bounded mean in [0,C] and additive \sigma-sub-Gaussian noise. It is confusing that P is used as an arbitrary transition kernel from X to Y while it has already been used in the MDP formulation. Then, in Assumption 3.1, at line 200, we have \mathcal{R}\subset \mathcal{P}_{S\times A, \mathbb{R}}^{g_R,\sigma_R} and  \mathcal{P}\subset \mathcal{P}_{S\times A, S}^{g_P,\sigma_P}. Here, the superscript and subscript interchanged. But the main issue is that the definition of the distribution family at line 198 implies that

\mathcal{P}_{S\times A, S}^{g_P,\sigma_P} = \{P(\cdot\mid s,a): (s,a)\in S\times A, \| E_{s'\in P(\cdot\mid s,a)}[s']\| \leq g_P(t), ...\}

Here, we have ||E[s']||. How can a state have a numerical value? We also have a similar issue at line 203 saying that \|E[s_{t+1}\mid s_t,a_t]\|_2 \leq g_P(t). Are the states embedded in some Euclidean space \mathbb{R}^d?


- M* appears at line 203 and used in the regret description but it is not clear what it means. At line 226, it is called "the original MDP M*", but it is not clear what original means in this context since only one MDP is described at line 162.

Assumption 3.1 and the problem formulation are critical for this paper as it introduces a new game class, called "hunting games". Therefore, a mathematically rigorous formulation is essential.

**Questions:**

- Can you provide a precise problem formulation addressing the issues highlighted in the weaknesses?

---

> ### Author Response · Authors · 2025-11-26
> **Response to Reviewer gYu8**
>
> We thank the reviewer for the insightful feedback and helpful suggestions. Please let us know if you have lingering questions and whether we can provide any additional clarifications during the discussion period to improve your rating of our paper. Below we address the specific questions and weaknesses pointed out by the reviewer.
>
> * The problem formulation is not clear. It starts with a standard MDP formulation but some "bounded growth" is assumed in Assumption 3.1. This would have implied time-varying MDP yet there is no discussion on that.
>
> This is not a time-varying MDP in the sense that the transition function changes over time. Instead this paper makes the case that the apparent operational regions in the state-space grows over time. In quite a large fraction of practical MDPs. This can make the RL algorithm convergence faster/slower. We take a deep dive to study these effects in this paper.
>
> * Reward function is defined as $R:S\times A  \rightarrow  (0,1]$ at line 163. However, at line 173, we have $r\sim R(s,a)$ as if it is a probability distribution.
>
> We thank the reviewer for pointing this out. We have edited it for consistency.
>
> * At line 195,   is introduced as the family of distributions ……How can a state have a numerical value?... . Are the states embedded in some Euclidean space ?
>
> We thank the reviewer for pointing out the notational inconsistencies. The updated document has these corrected. In short, yes, the states are embedded in a Euclidean space.
>
> * M* appears at line 203 and used in the regret description but it is not clear what it means. At line 226, it is called "the original MDP M*", but it is not clear what original means in this context since only one MDP is described at line 162.
>
> Line 162 describes a general MDP M. Since in this paper we have two distinct MDPs, an MDP of the actual system - M*, and the MDP that is estimated M_k, we pick this notation.
>
> * Can you provide a precise problem formulation addressing the issues highlighted in the weaknesses?
>
> The revised paper should give a more complete presentation of the problem statement. We have addressed the notational issues pointed out.  In short, the problem we address is the following : under conditions of bounded apparent state-space and reward growth given by the functions $g_T$ and $g_R$, how do we bound the regret when using a PSRL algorithm for finite horizon MDPs.

---

### Author Response · Authors · 2025-11-26
**Authors response to all the reviewers**

Dear Reviewers,

We thank you all for your insightful feedback and helpful suggestions.

We are grateful that reviewer yhQ4 and kCxi rated our soundness and presentation as good. We are happy to see quite a few positive comments on our paper.

We have addressed each weakness and question for all reviewers below their review. We have also made some minor changes to the paper and highlighted all changes in red in the updated pdf.

Please let us know if you have any lingering questions and whether we can provide additional clarifications during the discussion period to improve your rating of our paper.

Sincerely,
Authors

---

### Meta-Review · Area_Chair_xw2v · 2026-01-02

**Summary:**

This paper introduces a potentially novel problem setting in RL that the authors call "hunting game." This setting is supposed to capture environments where an agent is pushed into previously unseen regions of the state space due to the environment's heavy-tailed variability, even in the absence of adversarial dynamics. The paper offers a purely theoretical analysis of the proposed setting.

Some of the reviewers recognized novelty and potential interestingness of the proposed setting. Other reviewers pointed out lack of clarity of the paper in terms of: (i) motivation (i.e., lack of concrete grounding in practical RL problems that make the hunting game an interesting setting to study and not just a pure thought experiment), (ii) assumptions underlying theoretical analysis (e.g., contradiction between heavy-tailed setting of interest and sub-Gaussianity assumptions about the noise in the analysis), and (iii) somewhat convoluted style of writing.

For a paper that introduces a very novel setting, clarity of motivation and presentation of the work is absolutely paramount for the paper to have a chance for impact. Lack of clarity would certainly create confusion of any expert reader, which would likely result in the community dismissing this work altogether, nullifying its potential for impact. For this reason, I recommend rejecting the paper.

I appreciate the authors engaging in the discussion and suggest the authors focus on figuring out the best way to motivate, position, and present their work to maximize its impact, and resubmit the paper elsewhere.

**Reviewer Concerns:**

The major concern is lack of clarity of the paper along multiple axes:
- Lack of clarity in the motivation of the introduced setting (hunting games); the papers lacks a solid grounding in practical RL problems.
- Lack of clarity of some of the assumptions underlying theoretical analysis.
- Convoluted style of writing.

**Reviewer Scores:**

Based on the author's responses to reviewer's concerns, I don't believe any of the reviewer's would have increased their scores.
Reviewer YhQ4 might have lowered their initial positive rating of 8 given their confidence of 2 and weaknesses pointed out in other reviews.

---

### Decision · Program_Chairs · 2026-01-26

Reject